# Modeling interconnected minerals markets with multicommodity supply curves: examining the copper-cobalt-nickel system

John Ryter [1] ✉, Karan Bhuwalka[2], Richard Roth[3], Elsa Olivetti [4], Laura Buarque-Andrade[5], Max Frenzel [5], Ensieh Shojaeddini[1,6], Elisa Alonso[1] & Nedal Nassar [1]

Demand for many of the metals used in the energy transition is expected to grow rapidly. Many of these are by-products, often considered critical because their production responds weakly to prices and is instead tied to the economics of the host mineral. We present a model of prices and production for jointly produced commodities that accounts for interconnectivity between host and by-product markets at the mine level. We demonstrate this method using the copper−cobalt−nickel system, in which approximately 99% of cobalt is a by-product of copper or nickel mining. Our results show that the model more accurately captures the economic benefits of diversified mine outputs than previous approaches. Furthermore, changes in demand drivers for any two commodities produce non-linear effects on production and price. We challenge the prior best-practice assumption that cobalt cannot impact the copper or nickel markets. Recognizing the importance of both copper and cobalt for future electrification, we emphasize that incentivizing the copper industry to reduce cobalt supply risks could inadvertently undermine copper supply.

The demand for mineral commodities has risen alongside the need for technologies that utilize their unique properties[1]. The pursuit of higher energy density in batteries, lightweight and higher strength materials in transportation, smaller semiconductors, and enhanced efficiency in power conversion technologies, for example, has spurred interest in materials drawn from across the periodic table[2]. Mineral deposits typically contain a complex mix of commodities, and most metal mining projects involve co-product or by-product extraction[3]. Elements like cobalt, scandium, gallium, and neodymium provide desirable properties but provide only minority revenue shares at their respective mines, making their economic incentives less clear than those comprising majority revenue share like copper, iron, or aluminum[4–7].

This opacity, combined with the dependency of by-product production on the corresponding host commodities, has led to the use of by-product status as a criterion for material criticality classification in many studies[5,8,9]. Macro-scale studies have shown positive long-run correlations between host and by-product commodity production[10], a negative relationship between host and by-product commodity prices[4], and larger price volatility for by-product commodities than host metals[11], Others highlight the benefits of joint production, suggesting that producing multiple commodities can stabilize cyclical mining activity and stabilize both host and by-product prices[12]. With the high incidence of joint production and evidence for inter-commodity price and production relationships, understanding these inter-commodity relationships can aide in evaluating supply chain resilience and by-product status contribution to criticality.

Many modeling approaches rely on aggregated representations of mineral supply and demand[13–15]. While some of these frameworks

[1]U.S. Geological Survey, National Minerals Information Center, Reston, VA, USA. [2]Stanford University, Precourt Institute for Energy, Stanford, CA, USA. [3]Massachusetts Institute of Technology, Materials Systems Laboratory, Cambridge, MA, USA. [4]Massachusetts Institute of Technology, Department of Materials Science and Engineering, Cambridge, MA, USA. [5]Helmholtz-Zentrum Dresden-Rossendorf, Dresden, Germany. [6]Akima System Engineering, Contractor to the U.S. Geological Survey, Reston, VA, USA. ✉e-mail: jryter@usgs.gov

account for by-product behavior[7,15,16], they fail to capture the effects of changes at individual facilities, which is particularly important for many of the minor metals with relatively few producers. Consequently, mine-level representations may be more effective. While beyond the scope of this work, firm-level multi-commodity production optimization methods exist[17,18]. However, they rely on a variety of complex cost allocation methods, where method selection can substantively impact results[19]. Additional joint production models exist, but focus on production and recovery requirements to meet demand, neglecting mine economics[20].

Single-commodity mine-level modeling approaches often use supply curve-based methods, where relative production costs combine with demand to set price as the cost of the highest-cost producer[21–24]. Such methods are primarily used for commodities comprising the majority of mine revenues and has seen only limited extension to by-products. By-product supply curves have been proposed and used[22,25–27], but tend to function independent of host commodity economics or require allocating costs to the by-product commodity, a variable and uncertain process. One previous effort sought to use independent supply curves for copper, cobalt, and nickel production[22]. There, host commodity mine decision-making occurred independent of the by-product market and relied on revenue-based cost allocation to estimate prices. However, revenue fractions are based on the relative prices of the commodities, producing circular logic. Similarly, host, by-product, and co-product status are functions of the relative prices of the jointly produced commodities, where sufficiently large price changes could alter the commodity used for mine decision-making. The methods described often overlook inter-commodity interactions and feedback, limiting their effectiveness in understanding market dynamics for jointly produced commodities.

The method developed here addresses these challenges by considering the revenue contributions of all jointly produced commodities simultaneously, relying solely on total production costs to avoid commodity-cost allocation issues. Rather than using two-dimensional supply and demand curves, this method demonstrates that for a two-commodity system, three-dimensional supply and demand surfaces can be constructed for each commodity as functions of both commodities' prices. Solving for supply-demand equilibrium across both commodities simultaneously produces a single, unique combination of prices. For a three-commodity system, supply and demand for each commodity are functions of all three commodities' prices, requiring a four-dimensional approach, where a unique price combination exists that satisfies all three commodities' equilibria. Neither three-dimensional nor four-dimensional approaches to modeling mineral production and price have been documented to date, and this method represents a substantive contribution to the literature in its generalization to consider any number of commodities simultaneously, with capacity to represent any jointly produced commodities.

The authors have previously attempted generalized mineral supply chain modeling for host commodities in isolation[28]. In place of a supply curve-based approach, price-responsive reserve to production ratios were used to generate simulated mines available for opening, and the probability of mine opening was tuned to match historical production. Where this previous work was used to understand how elasticities and supply chain response rates differ across host mineral commodities, the method presented in this work could permit more robust forecasting informed by multiple future policy scenarios with improved model stability. This method also reduces data requirements and the model complexity per commodity while enabling the representation of multiple commodities simultaneously.

We demonstrate the utility of this method using cobalt, copper, and nickel as a three-commodity interconnected system. In 2022, ~1% of cobalt was produced from cobalt-primary mines, with 74% as a co-product of copper mines and 25% as a by-product of nickel mines[29].

Although cobalt revenues are minor compared to copper and nickel, they still influence mine profitability. On the demand-side, cobalt, copper and nickel all have experienced the impact of the clean energy transition, although that impact has varied. For example, lithium-ion batteries have become the dominant end-use for cobalt given the rapid global growth of demand for electric vehicles and energy storage systems. Lithium-ion batteries are one of many nickel end-uses, but not yet one of the dominant applications. On the other hand, copper demand in the clean energy transition is tied to its use as an electrical conductor, where it is needed across a broad range of applications such as electric vehicles, charging stations, and electricity grids.

These supply-demand factors, alongside market responsiveness delays, create complex price dynamics that necessitate a comprehensive modeling approach[30]. Although the potential challenges of motivating increased production of by-product commodities are well-described, this method provides a quantitative tool for policymakers and market participants seeking better insights into future by-product supply and demand dynamics. The model could be integrated with detailed demand scenarios such as those from the International Energy Agency[31] or the Energy Information Administration[32]. The following section presents modeling results with theoretical demand scenarios that highlight the detailed insights needed to understand the role of multicommodity production on historical and future supply.

## Results
### Historical tuning accuracy
A short description of the method is provided here to provide context for the initial results. With a dataset on mines, requiring only metal production, total ore treated, and cost per tonne (metric ton) of ore treated[33], and rules for their opening, closing, and operation, our method was applied to simulate annual production and price for each commodity simultaneously, starting in 2001 to enable tuning and validation on historical data. For the three-commodity system of copper, cobalt, and nickel, the method estimates supply and demand for each commodity as a function of all three commodities' prices, producing four-dimensional (4D) supply and demand functions. Solving for the intersections of the 4D supply and demand functions produces a surface for each commodity, representing all points of supply-demand equilibrium for that commodity as a function of all three commodities' prices. The point where these three equilibrium surfaces intersect gives the common point of supply-demand equilibrium across all three commodities, and therefore the corresponding equilibrium prices of each commodity. Finally, plugging these prices back into the original supply and demand functions for each commodity gives the corresponding supply of each commodity, as well as the mines that open and close to satisfy it. This method is described in detail in the Methods section and is referred to as the 4D approach due to its use of 4D supply and demand functions.

To compare the 4D method with the most advanced prior approach using the same input data, the method used by Nguyen et al. was implemented as well[22]. Here, the intersection between the two-dimensional (2D) nickel supply and demand curves, where supply and demand are functions only of the nickel price, was solved to find the equilibrium nickel price. Treating this nickel price as fixed, the process was repeated using the 2D copper supply and demand curves to find the equilibrium copper price, and with fixed copper and nickel prices the process was repeated for cobalt. This approach is also described in further detail in the Methods section and is referred to as the 2D approach due to its use of 2D supply and demand functions.

With mine costs and production quantities given, the supply curve is generally described by the sum of commodity production for all mines profitable at a given price combination. The demand curve is described by Eq. 1, using the general form reflecting the inverse correlation between price and demand[34], and corresponding with the

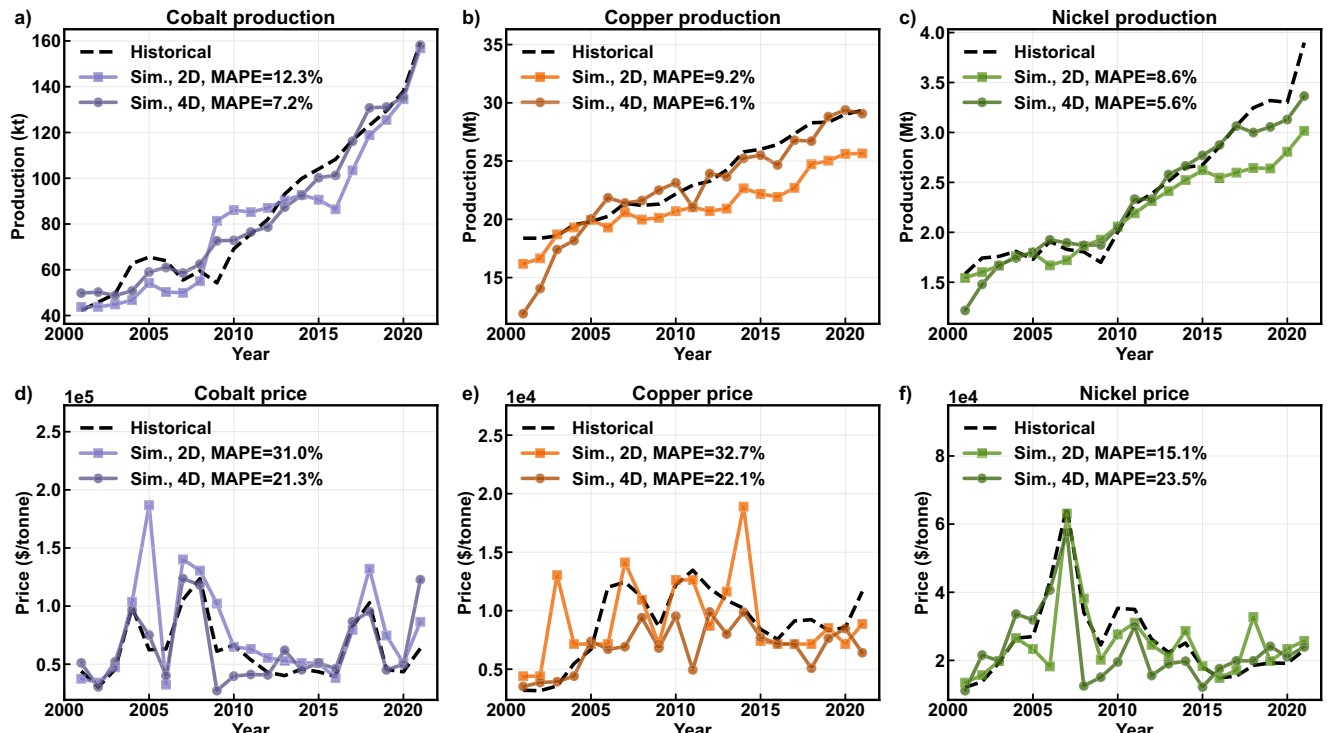

**Fig. 1 | Historical and simulated production and price.** Actual and simulated cobalt (**a**), copper (**b**), and nickel (**c**) production and cobalt (**d**), copper (**e**), and nickel (**f**) price for the 2D and 4D approaches. Simulated values are from historical modeling 2001–2021, with the mean absolute percentage error (MAPE) shown in the legends of the corresponding plots. Prices are adjusted for inflation. $/tonne = 2023 US Dollar per metric ton; kt = thousand metric tons; MAPE = mean absolute percentage error; Sim. = simulated.

derivation of the price elasticities of demand used in this work[35].

$$\text{price}_{c,y} = \alpha_{c,y} \left( \text{demand}_{c,y} \right)^{1/\beta_c} \tag{1}$$

Where $\beta_c$ is the is the price elasticity of demand for commodity $c$ and $\alpha_{c,y}$ is the demand shifter for commodity $c$ in year $y$. The elasticity is constant over time, while the demand shifter changes as a function of underlying demand drivers, such as gross domestic product (GDP) or electric vehicle demand. The value of the demand shifter for each commodity in each year was determined separately for the 2D and 4D approaches using Bayesian optimization, where the objective was to minimize the difference between historical and simulated production and price. The results of this process are shown in Fig. 1.

The method is capable of reproducing historical data with an accuracy comparable to or exceeding that of previous methods, which do not permit multidirectional inter-commodity interactions. MAPE values associated with this work are shown in Fig. 1 alongside general simulation results. For comparison, the Cu-Ni-Co method developed by Nguyen et al[22]. produced historical validation MAPE values of 5.1, 7.2, and 7.1% for production and 48.8, 27.9, and 19.1% for price, for cobalt, copper, and nickel, respectively (Supplementary Table 2). MAPE values for production are generally comparable, as are those for host-mineral commodity prices. However, this work shows substantial improvements in reproducing cobalt prices, does not require an initialization period, and considers multidirectional interactions between commodities.

**Commodity interactions**

To understand how one commodity affects another within the 2D and 4D regimes, the consequences of varying the demand shifters for individual commodities were evaluated from 2022 to 2033, the last year of potential new mine data. This approach allows for simplified theoretical demand scenarios rather than selecting from numerous forecasts. Increasing the demand shifter moves the theoretical demand curve up and right in a single-commodity model with constant price elasticity. In the 4D regime, the entire demand function shifts toward higher own-commodity price and production values.

Five demand shifter pathways were evaluated for each commodity, with slopes defined as of 100%, 50%, 0%, −25%, and −50% of the 15-year average slope of the tuned, base-10 logged demand shifter. Demand shifters are large due to small price elasticities of demand for copper (−0.05), nickel (−0.09), and cobalt (−0.45) and the formulation of Eq. 1 (Supplementary Section 1.12). The asymmetrical slope fractions reflect expected future growth in copper, cobalt, and nickel demand drivers, with a maximum of 100% due to the likelihood of missing future supply in the dataset. Each pathway represents changes in the demand shifter over time relative to the tuned 2021 value. All possible combinations of these five slopes were tested, giving 125 (5³) scenarios. Fig. 2a–c illustrates these pathways in the 4D regime, while Fig. 2d–i shows the corresponding production and price trajectories. Although these figures highlight several scenarios and show the range across all scenarios, they do not capture inter-commodity interactions or differences between the 2D and 4D methods. However, cobalt does exhibit a larger number of production and price pathways than copper or nickel, attributable to stronger nickel and cobalt demand shifter effects on cobalt production.

To highlight inter-commodity effects, marginal price and production responses were calculated for each commodity-growth rate combination. The marginal response is defined as the change in production or price resulting from a demand shifter slope changing for an individual commodity relative to the zero-change scenario, holding other commodities' demand shifter growth rates constant. For example, the marginal responses for the cobalt demand shifter at 100% of the historical slope are each defined as $p_{d_{Co,x}, d_{Cu,y}, d_{Ni,z}} - p_{d_{Co,0}, d_{Cu,y}, d_{Ni,z}}$, where $p$ is the time series vector for price or production associated

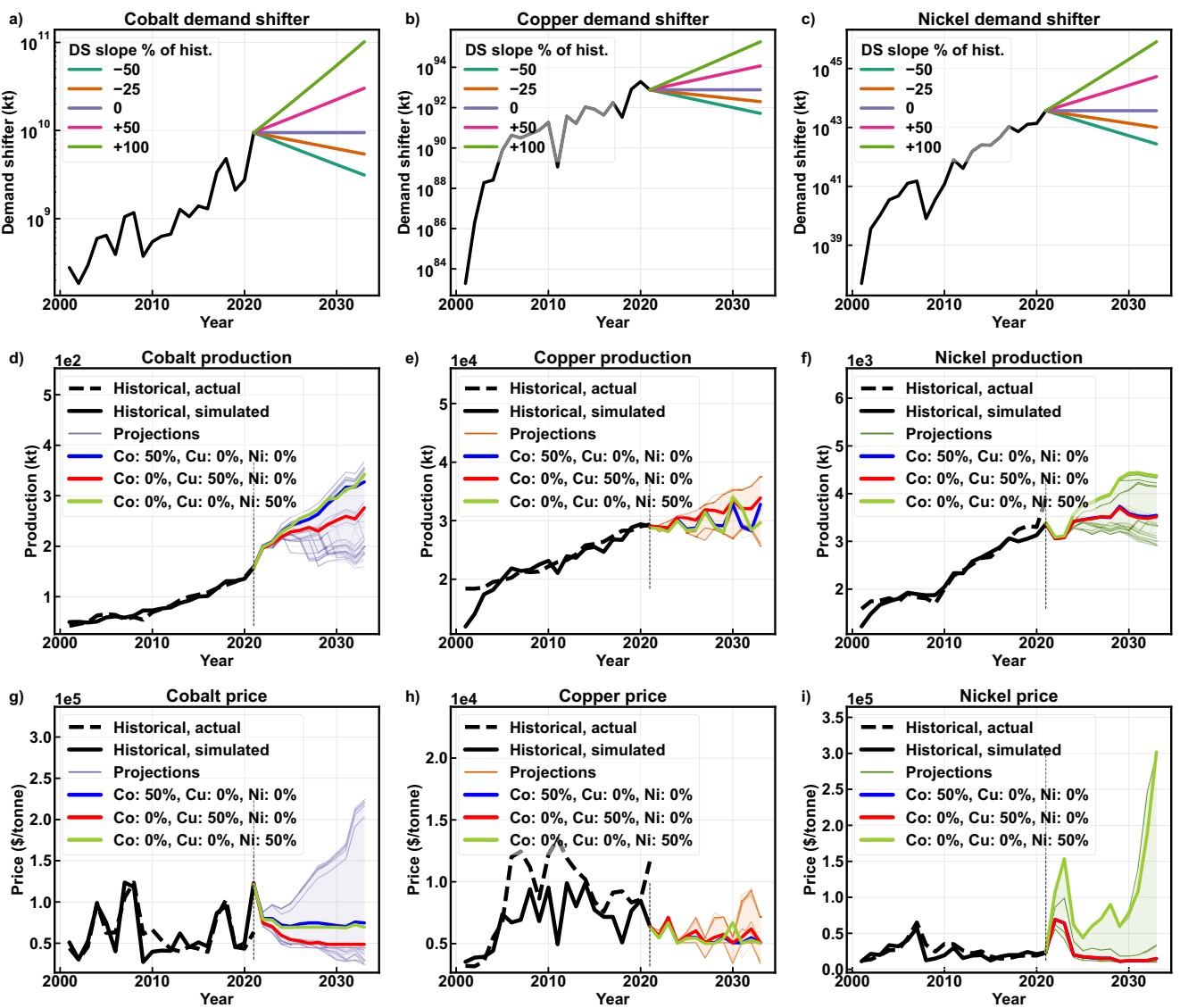

**Fig. 2 | Demand shifter pathways, with resulting production and price forecasts.** Historical and experimental demand shifters for (**a**) cobalt, **b** copper, and **c** nickel. Resulting production (primary + secondary) for (**d**) cobalt, **e** copper, and **f** nickel. Resulting prices for (**g**) cobalt, **h** copper, and **i** nickel. Shading represents the 5th to 95th percentiles for each year, and the vertical dashed lines indicate the transition from historical to projected values in 2021. The three individual scenarios associated with 50% demand shifter growth for each commodity are highlighted in d-i to provide examples of scenario pathways across subfigures, with demand shifter growth rates for each commodity shown in the respective legends. $ = 2023 U.S. dollar, DS = demand shifter, kt = thousand metric tons.

with a combination of demand shifter slopes $d_{Co,x}$, $d_{Cu,y}$, and $d_{Ni,z}$ with $x = 100$, while $y$ and $z$ can take any values in (−50, −25, 0, 50, 100). Marginal production and price were calculated for both the 2D and 4D approaches and are presented in Fig. 3 (production) and 4 (price). Each commodity-growth rate combination yields 25 marginal responses based on all combinations of the other two commodities' demand shifter growth rates.

Figs. 3 and 4 enable comparison of model results when inter-commodity relationships are fully interconnected (4D) and when mine decision-making flows unidirectionally from the host commodity (2D). The overall range of own-commodity effects for production and price are similar between the 2D and 4D cases (Fig. 3a, e, and i; and Fig. 4a, e, and i) with the exception of copper price. The copper price exception is due to differing 2021 demand shifter values between the 2D and 4D cases, and the resulting mines available for opening. Cobalt and nickel also exhibit extreme own-commodity price effects (Fig. 4a, i), with 2033 values exceeding 300% and 1000% respectively. The magnitude of these values is due in part to the low prices for the reference case,

where the demand shifter remains constant at 2021 levels, but may also be due to more limited future mine data, particularly for nickel.

Observing individual marginal production changes, such as nickel production with 100% nickel demand shifter slope (Fig. 3i, purple), demonstrates a greater number of evolution pathways in the 4D case than the 2D case. This variation indicates that demand shifter effects are not strictly linear in the 4D case, indicating production for one commodity also depends on the values of other commodities' demand shifters – meaning nickel production responds differently to changes in nickel demand shifters depending on the rate of change for cobalt demand shifters. This result suggests that future changes in demand shifters, such as electric vehicle battery demand, which affect demand for both nickel and cobalt, can have compounding effects on both nickel and cobalt production.

In the 4D case, changes in the cobalt demand shifter can produce substantial effects on production and price for copper (Figs. 3b and 4b) and nickel (Figs. 3c and 4c), while the effects are limited in the 2D case. The copper and nickel markets are capable of influencing each

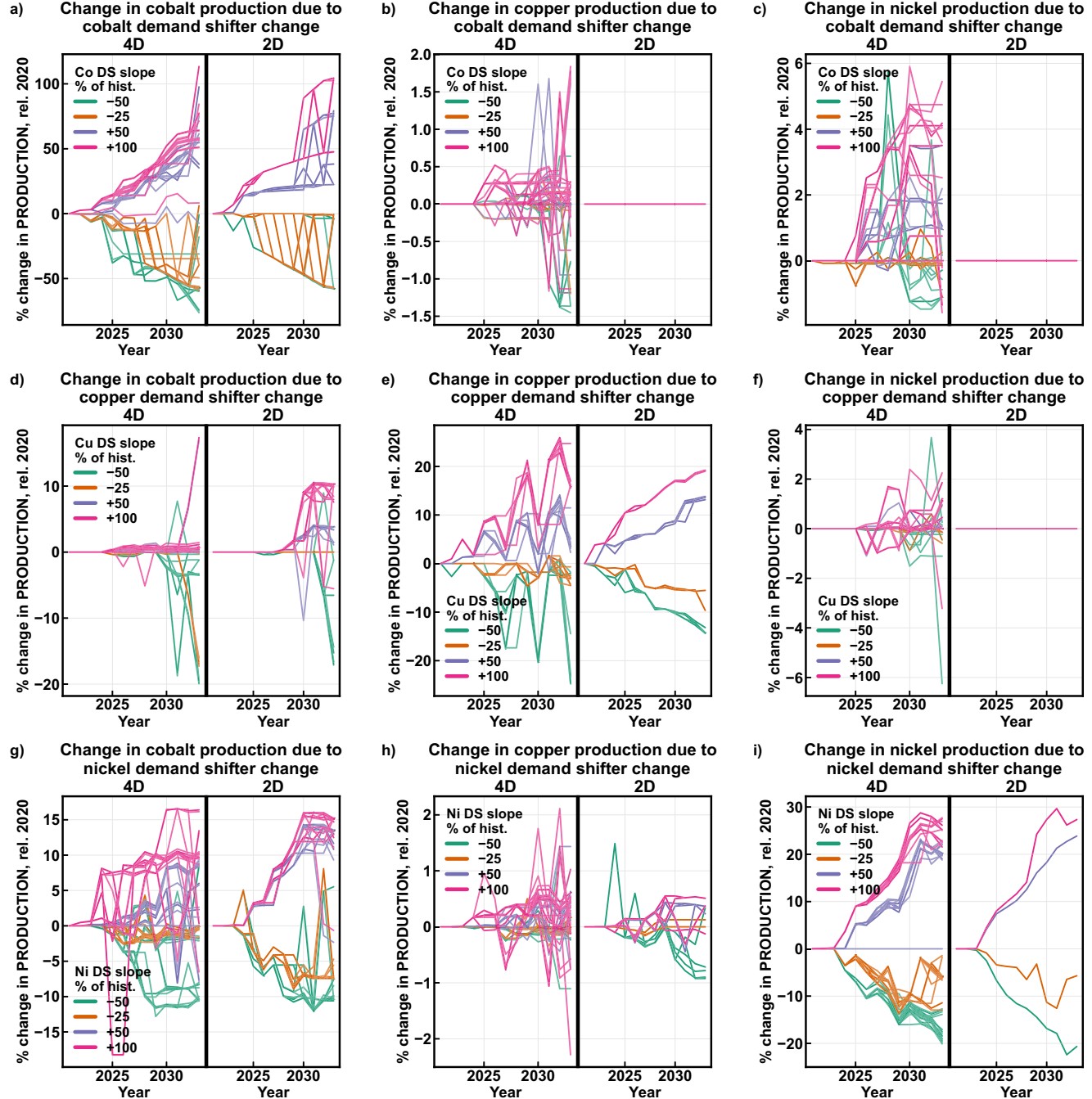

**Fig. 3 | Changes in production based on changes in demand shifters.** For each scenario, the change in production for each mineral commodity is compared to the change in demand shifter (DS) for each mineral commodity: change in (**a**) cobalt production for cobalt DS; **b** copper production for cobalt DS; **c** nickel production for cobalt DS; **d** cobalt production for copper DS; **e** copper production for copper DS; **f** nickel production for copper DS; **g** cobalt production for nickel DS; **h** copper production for nickel DS; **i** nickel production for nickel DS. The model is run in two modes: the 4D mode considers by-product/co-product effects on the host commodities while the 2D mode does not. DS = demand shifter, hist. = historical, rel. = relative to.

other as well (Figs. 3f, h and 4f, h). While the effects are relatively limited, they challenge the notion that by-product and co-product commodities cannot influence mine decision making. Cobalt price is more resilient to fluctuations in copper and nickel demand shifters in the 4D case, likely resulting from the additional degrees of freedom in supply and better representation of economy of scope. Copper production and price respond to changes in the nickel demand shifter in the 2D case (Figs. 3h and 4h) because the nickel supply curve is solved before the copper supply curve in this implementation of 2D approach, resulting in copper by-product behavior at nickel-copper

mines. This behavior is not present in other studies' implementations of the 2D method because they treat copper and nickel production as entirely independent, which was not possible using this dataset.

These results demonstrate a characteristic of supply and demand curve relationships across multiple commodities. Considering the demand shifter as a measure of willingness to pay, economic theory indicates that increasing the host commodity willingness to pay should increase the host commodity price and production, while also increasing by-product commodity production and thus decreasing by-product commodity price, all else equal. This work validates these

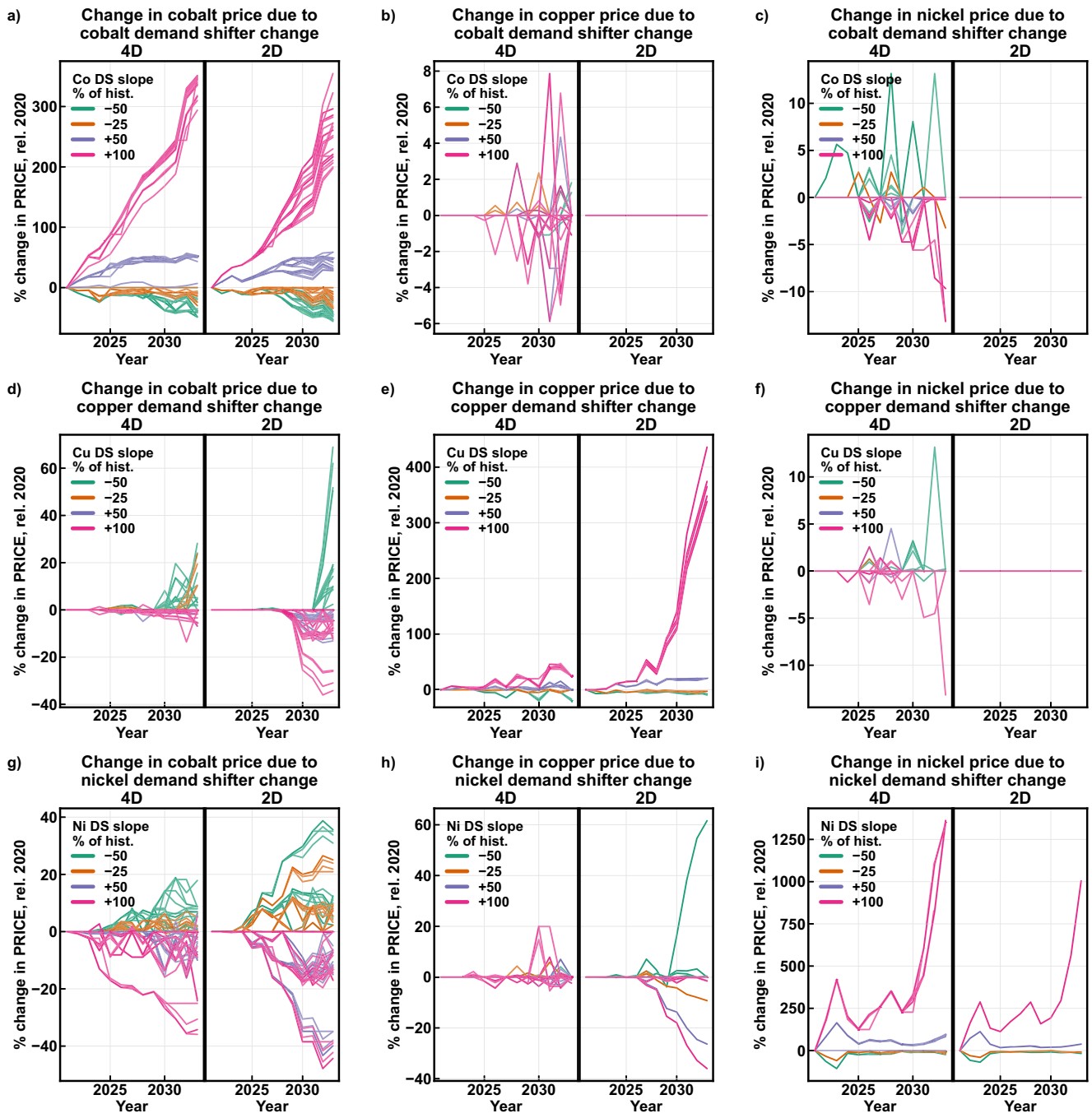

**Fig. 4 | Changes in prices based on changes in demand shifters.** For each scenario, the change in price for each mineral commodity is compared to the change in demand shifter (DS) for each mineral commodity: change in (**a**) cobalt price for cobalt DS; **b** copper price for cobalt DS; **c** nickel price for cobalt DS; **d** cobalt price for copper DS; **e** copper price for copper DS; **f** nickel price for copper DS; **g** cobalt price for nickel DS; **h** copper price for nickel DS; **i** nickel price for nickel DS. The model is run in two modes: the 4D mode considers by-product/co-product effects the host commodities while the 2D mode does not. DS = demand shifter, hist. = historical, rel. = relative to.

expectations and expands on the theory above: increasing the copper demand shifter raises its demand curve, producing an increase in copper production (Fig. 3e) and an increase in copper price (Fig. 4e). This increase also raises cobalt production (Fig. 3d), but produces a decrease in cobalt price (Fig. 4d). The decreasing cobalt price is noteworthy and has not been previously explained, despite occurring in both the 2D and 4D cases. An increase in the copper demand shifter raises copper price and production, increasing copper revenues for all mines producing it and acting as a credit for copper-cobalt mines, lowering the cobalt price needed for positive free cash flow. All

copper-cobalt mines shift lower on the cobalt supply curve, enabling higher cobalt production and lower price for the same cobalt demand curve. The nickel demand shifter produces similar cobalt outcomes (Figs. 3g and 4g).

## Discussion

This work introduces a method for simultaneously modeling multiple jointly produced mineral commodities' prices, supply, and demand, considering multidirectional inter-commodity relationships. We compare this method with the previously most advanced method and find

this method offers several advantages, including better reproduction of historical price and production for nearly all cases. It enables explicit representation of individual mines and their behaviors using a limited dataset: annual commodity production, cost per tonne of ore treated, and quantity of ore treated. Coupled with detailed multi-commodity data on potential deposits, this approach allows for forward-looking supply projections without needing to exogenously determine by-product or co-product status, or perform potentially problematic cost allocations, of jointly produced mineral commodities. The model can also assess various demand growth trajectories for different commodities while endogenously estimating their price structures and interactions. Understanding future mineral commodity prices can aide in evaluating the affordability of low-carbon energy and mobility technologies, their rates of adoption, and the mining operations necessary to fulfill the associated mineral demand.

Additionally, this method challenges the underlying assumptions of prior mineral supply models, namely that by-product and co-product commodities do not influence host commodity mine decision making. Not only does this method highlight substantial by-product / co-product effects on production and price for copper-cobalt and nickel-cobalt mines, but also for copper-nickel and nickel-copper mines. Game theory approaches have found that joint production enables implicit interactions even between single-commodity firms, such as copper and cobalt recyclers, due to their competition with multi-commodity firms[36], providing justification for these findings. Similarly, econometric studies of jointly-produced commodities' prices indicate host commodity price effects on by-product commodity prices have been overestimated, and that by-product price effects on host commodity prices have been underestimated[37]. Furthermore, this method demonstrates that cobalt prices are more robust to host-commodity demand shifts when multidirectional relationships are considered, based on cobalt price response to copper and nickel demand shifters in the 2D and 4D cases. This result suggests this method better captures economy of scope, or the increased economic competitiveness for firms with more diversified outputs[38].

This method also indicates that changes in demand drivers for any two commodities produce nonlinear effects on long-term production and price trajectories. For example, nickel production's response to changes in nickel demand shifters varies with the rate of change for cobalt demand shifters. These results indicate that as energy transition-induced demand for minerals rises, the prices and production of jointly produced commodities may become increasingly interconnected. With social concerns and policies incentivizing the copper industry to reduce cobalt supply risks, this approach indicates that despite cobalt-producing mines accounting for only ~1% of copper production[39], efforts to influence the supply or demand of cobalt could negatively impact the copper market. As a result, host commodity markets may require additional support or investment when policies intending to affect their by-products' markets are implemented.

A key limitation of forward-looking models is that results are substantially affected by the current knowledge of the supply pipeline. The model addresses historical production gaps via resampling and imputes costs for potential new mines lacking data. For mines projected to come online in the farther future, production estimates, timelines, and imputed costs become increasingly uncertain. Moreover, some production is occurring in countries where reporting requirements are lax or where new mine development is outpacing reporting capabilities. Variations in data quality between commodities could also explain some model observations, such as nickel's high price responsiveness relative to that of copper. Similarly, although this work accounts for revenue contributions of other jointly produced commodities at all mines using static prices and production, the presence of substantial gold, silver, molybdenum, and other commodity

production at copper mines may dampen their price responsiveness relative to nickel. Expanding model dimensionality to account for these commodities is a potential avenue for future work.

Several simplifications were made to focus on method development. The model relies on average cost curves, which identify the marginal producer as the basis for price setting, because marginal costs are not reported. Incorporating mine capacity expansion and production adjustments would increase realism but require additional cost data. For new mines, the current approach requires positive free cash flow in the opening year, neglecting capital expenditures and construction timelines. A more realistic approach would require that each new mine's net present value evaluated over its lifetime exceeds zero based on some project-dependent discount rate, which would increase the prices required for mine opening relative to the current free cash flow approach and would necessitate data on construction costs.

This model utilizes some of the best available public and commercial datasets, and adding further cost details is particularly challenging given varying transparency and disclosure practices throughout the industry. The method can be adapted to permit usage of fundamental data reported in company filings such as annual capital and operating expenditures, negating the need to differentiate between different cost bases, e.g. C1, C3, copper-equivalent[40,41]. However, data gaps may require additional modeling to estimate costs, ore grades, and other mine information[42]. Similarly, investments in plant upgrades, recovery circuits, process plant re-optimization and other actions that change joint production ratios and marginal costs are challenging from both data and modeling perspectives. However, this method's implementation, where mine production is calculated as a function of each price combination, would be compatible with more complex mine production optimization algorithms, if data were available. However, mineral concentrate prices, treatment charges, and refining charges also vary based on long-term contracts and the presence of payable and penalty elements. While the current approach contains static representations of treatment charges and refining charges (TCRCs), TCRCs fluctuate with supply-demand imbalances in the mineral concentrate market[43]. Refining also has the capacity to act as a bottleneck in mineral supply[44], and representing it explicitly would improve the model.

Future applications of this method could develop detailed forward-looking scenarios, including for other commodity systems, particularly where joint consumption or material substitution introduces cross-price elasticities of demand between commodities. Several studies have found evidence that joint consumption can be a strong driver of inter-commodity price interactions[45,46], highlighting the need for explicit demand curve interdependencies. Furthermore, while this work uses simplistic commodity-agnostic demand shifter growth rates to study inter-commodity interactions, each commodity's demand is expected to grow with varying rates and volatilities[47]. To enable such forecasts, demand shifters can be represented following the same approach as econometric methods for elasticity estimation, where exogenous parameters like GDP, steel production, electricity grid expansion, and electric vehicle adoption rates can be used to model demand shifter values, enabling more complex commodity-specific forecasts. Spillover effects between host-byproduct pairs and their demand drivers may also need consideration to handle dynamic, multi-year impacts of demand shifter changes[48]. Additionally, non-economic factors such as $CO_2$ emissions and pricing could be incorporated alongside existing price axes to model green premiums and regional supply chain constraints. With the existing model, future applications could also include scenario analyses, where the mine-level model structure could permit studies on the effects of ore grade or TCRC variation, changes in investment risk tolerance, or country-level policies such as subsidies, reductions in permitting timelines, and export restrictions.

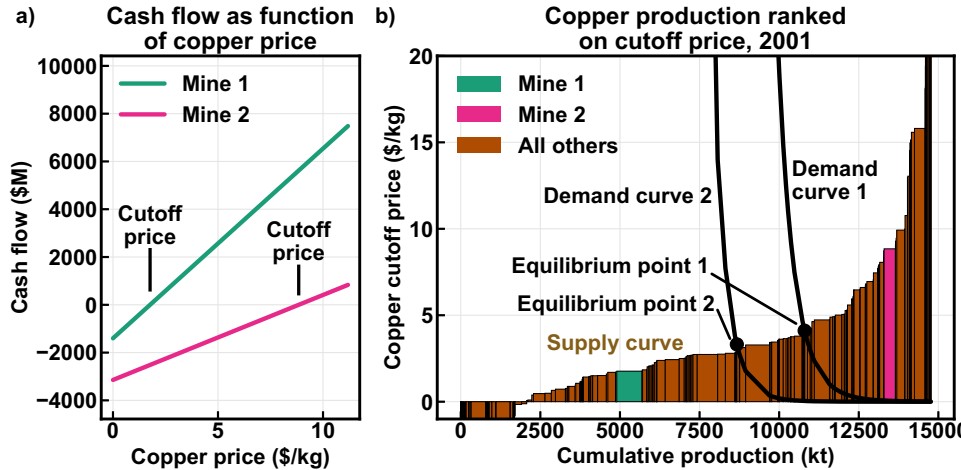

**Fig. 5 | The transition from cash flow to supply curve. a** Free cash flow as a function of copper price for two generic mine sites. **b** Copper cost curve and demand curve for the year 2001. Cutoff price values for several mines with extreme values are not displayed in full; negative values are due to revenues from other commodities. $M = million US dollars; $/kg = U.S. dollar per kg; kt = thousand metric tons.

## Methods

### Single-commodity approach

Typical approaches to modeling mine planning seek to maximize net present value (NPV, Eq. 2), where the discounted future free cash flows are given for a single-commodity mine in the simplified Eq. 3 [49]. Here, the term "simplified" indicates that there can be additional components to the production and cost variables, as paid metal production may be more suitable than production itself, and total cash cost can include market-determined costs like treatment and refining charges as well as variable costs such as labor. NPV is particularly useful for evaluating mine opening decisions, but free cash flows as defined in Eq. 3 rely on the total cash cost, described as the most useful measure of short-term mine profitability, despite excluding capital expenditures[50].

$$\text{NPV} = \sum_{t=0}^{T} \text{capital expenditure}_t + \frac{\text{free cash flow}_t}{(1+\delta)^t} \quad (2)$$

$$\text{free cash flow}_t = \text{production}_t \cdot \text{price}_t - \text{ore treated}_t \cdot \text{total cash cost}_t \quad (3)$$

Where $t$ is the time index starting from present, $T$ is the time horizon considered, and $\delta$ is the discount rate as a decimal value. When capital expenditures, production, total cash costs, and ore treated are fixed, the free cash flow may be plotted as a linear function of price as shown in Fig. 1a, using copper as an example commodity. The slope and intercept of this line vary with the deposit or mine site cost profiles. Assuming constant prices with time, NPV would show the same general behavior as free cash flow. Free cash flow was used rather than NPV throughout this study for simplicity, and all costs and prices were adjusted for inflation to 2023 U.S. dollars.

The intersection of the lines in Fig. 5a with zero gives the cutoff price for short-run mine site profitability. By sorting all available mines by their cutoff prices, the supply curve can be constructed, as shown for copper in Fig. 5b. Supply curves model mine production via opening and closure[26,27]. The intersection of the supply and demand curves indicates the price at which total demand equals total supply, determining the commodity price for that year. Mines with cutoff prices below this price can operate profitably, representing the marginal cost of production. Fig. 5b illustrates how different demand constants affect both production and price.

While the method described above has been used extensively in modeling many major mineral commodities[21–23], its utility is limited to commodities comprising the majority of the respective mines' revenues. However, this approach has been used to model the copper-cobalt-nickel system, and as the most advanced prior modeling approach, it was implemented here as described in Supplementary Section 1.2.

### Theoretical framework for a two-commodity system

When we expand to a multi-commodity mine, we add an additional dimension to each of the production, price, and cost variables in the free cash flow and NPV equations for each additional commodity initially defined only for the main commodity produced by the mine. This can be described by the multi-commodity Eq. 4.

$$\text{free cash flow}_t = \left( \sum_{c=0}^{C} \text{production}_{t,c}\, \text{price}_{t,c} \right) - \text{ore treated}_t\, \text{total cash cost}_t \quad (4)$$

Where $c$ is the commodity index and $C$ is the number of commodities produced at the mine. The total cash cost is per tonne of ore and excludes any by-product credits. Now, the cutoff price for profitable production at a mine site for one commodity becomes a function of the prices of all other commodities, with the extent of that dependence determined by the relative revenue fraction of each commodity. This cutoff price can be calculated as in Eq. 5, where $i$ represents the index of the commodity of interest.

$$\text{cutoff price}_{t,i}$$
$$= \frac{\text{ore treated}_t\, \text{total cash cost}_t - \sum_{c=0;\, c \neq i}^{C} \text{production}_{t,c}\, \text{price}_{t,c}}{\text{production}_{t,i}} \quad (5)$$

For a mine producing two commodities, such as copper and cobalt in the examples below, this increase in dimensionality produces a free cash flow plane rather than a line, where free cash flow is a function of the prices of both copper and cobalt. Here, given a higher cobalt price, the cutoff price for copper is lower, and vice versa. Like the single-commodity case, the slopes and intercepts vary with the mine site cost profile, but in both dimensions and also with the revenue fractions of each commodity.

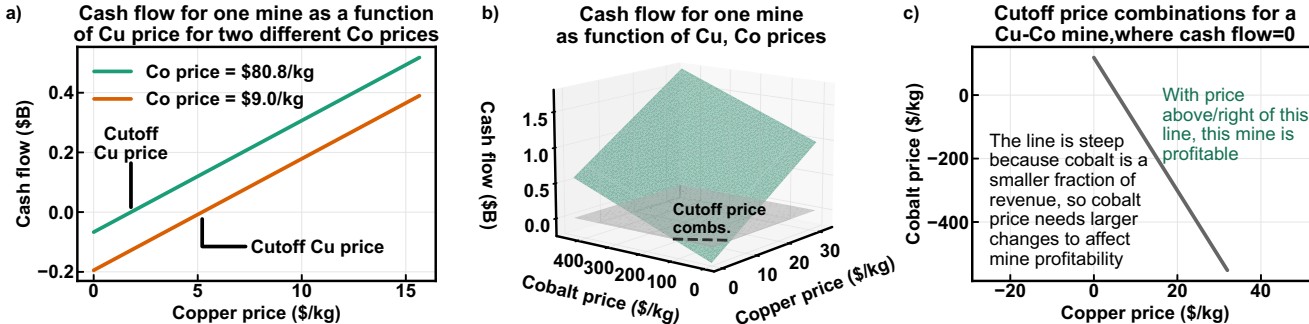

**Fig. 6 | Developing the basis for two-commodity supply surfaces. a** Free cash flow as a function of copper price for a single Cu-Co mine, given two different cobalt prices. **b** Cash flow for the same mine as a function of both copper and cobalt price, where the gray plane represents free cash flow = 0 and the dashed line represents the intersection of the two planes. **c** For the same mine, all breakeven price combinations that produce free cash flow = 0. Prices are in US dollars per kilogram. Combs. = combinations; $B = billion US dollars.

As above, the intersection of each mine site's free cash flow plane with the zero plane gives pairs of breakeven prices required for profitability. Rather than a single price for a single-commodity mine, the intersection of two planes produces a line highlighting the tradeoff between copper and cobalt prices, as shown in Fig. 6c. The line has a negative slope because as the copper price rises, a lower cobalt price is required to maintain equivalent profitability. The slope of the line is equivalent to the negative ratio of copper to cobalt production; as a result, larger changes in cobalt price compared to copper price are required to affect mine profitability. Similarly, relatively small copper price changes have a greater impact on mine profitability.

Fig. 6 illustrates how a mine's free cash flow is depends on its products' prices. Free cash flow can be calculated for any price combination, producing a free cash flow plane. Assuming mines operate when free cash flow is positive, different price combinations produce different combinations of operating mines. The total supply for any price combination is then the sum of production for all mines with positive free cash flow. This approach parallels the supply curve approach described above, but provides supply for two commodities as functions of both their prices simultaneously. Plotting copper and cobalt supply against their prices produces the supply surfaces shown in Fig. 7a and b. A typical single-commodity supply curve can be generated for each commodity by fixing the price of the other commodity.

In the single-commodity case, the intersection of the supply and demand curves determines the supply and commodity price. With two commodities, there are now two demand surfaces, and each intersects with a separate (though not independent) supply surface. Assuming that each commodity's demand is independent, each demand surface is simply a two-dimensional demand curve extended into the other commodity's price dimension. The intersection of each supply-demand surface pair produces a curve that varies along all three axes, as shown in Fig. 7c and d. These curves represent supply-demand equilibria as functions of both copper and cobalt prices.

For supply-demand equilibrium to occur for both copper and cobalt simultaneously, their prices must be constrained to price pairs residing on both curves. Projecting these curves onto the price plane (defined by $Supply_{Cu} = Supply_{Co} = 0$), reveals the intersection of these curves as the only point where both commodities' supply-demand equilibria are satisfied (Fig. 7e). This point represents supply-demand-price equilibrium, and the corresponding production values can be plotted on the supply surfaces (Fig. 7f, g). The algorithm for calculating this equilibrium point is detailed in Supplementary Section 1.12.

While the one-commodity case produces two-dimensional supply and demand curves, the two-commodity case yields three-dimensional surfaces. Extending the method to three commodities requires four

dimensions, visualized using a unique three-dimensional supply-surface for each value of the third commodity's price. The method is described in detail in Supplementary Section 1.1.

## Primary and secondary supply
Both primary and secondary supply were modeled in this work. For primary supply, the main data source for mine production and costs is the S&P Capital IQ Pro Metals and Mining Properties Screener[39], supplemented by feasibility study data in the S&P Capital IQ Pro Capital Costs Screener[33], and mine-level data from Gulley[51], Project Blue[52,53], and WoodMackenzie[54]. Data were imported for all mines producing copper, nickel, or cobalt in any year, for the years 1991-2040. Forward-looking data enables inclusion of mines forecasted to come online, but may not cover all mines under development and may underestimate future supply. For additional detail, refer to Supplementary Section 1.4. Secondary supply was modeled following the approach in Nguyen et al.[22], with additional detail in Supplementary Section 1.10. Mine-level behavior is detailed in Supplementary Section 1.9.

## Macro-scale data sources
Data for total demand, mine production, and secondary supply was calculated by taking the median value for each year across all data sources listed in Table 1, with resulting values shown in Supplementary Fig. 3. Secondary supply information was the least reliably available, and can represent either the estimated secondary recovery or secondary demand. Given that reporting of secondary supply and demand is variable, secondary supply and consumption were assumed equivalent to the maximum scrap supply available. The refined metal prices were from S&P Capital IQ Pro[33], and were adjusted for inflation as described in the Supplementary Section 1.8.

## Historical tuning
The annual supply curve is constrained by the mines available, while the constant demand elasticity over time leaves the demand shifter $\alpha_{c,y}$ (Eq. 1) as the only unconstrained variable in the system. The demand shifter, representing willingness to pay, changes over time and was tuned for each year using Bayesian optimization. This tuning aligns historical prices and demand with model outputs, providing baseline demand shifter values for comparison with historical demand drivers.

Because matching price and production simultaneously produces a multi-objective optimization problem with no objectively optimal tradeoff, the parameter $w_{\text{price tuning}}$ was introduced to allow for different weightings between price and production. Different $w_{\text{price tuning}}$ values generate the Pareto front[55], and the final value of $w_{\text{price tuning}}$ was selected as described in Supplementary Section 1.13: 0.086 for the 4D

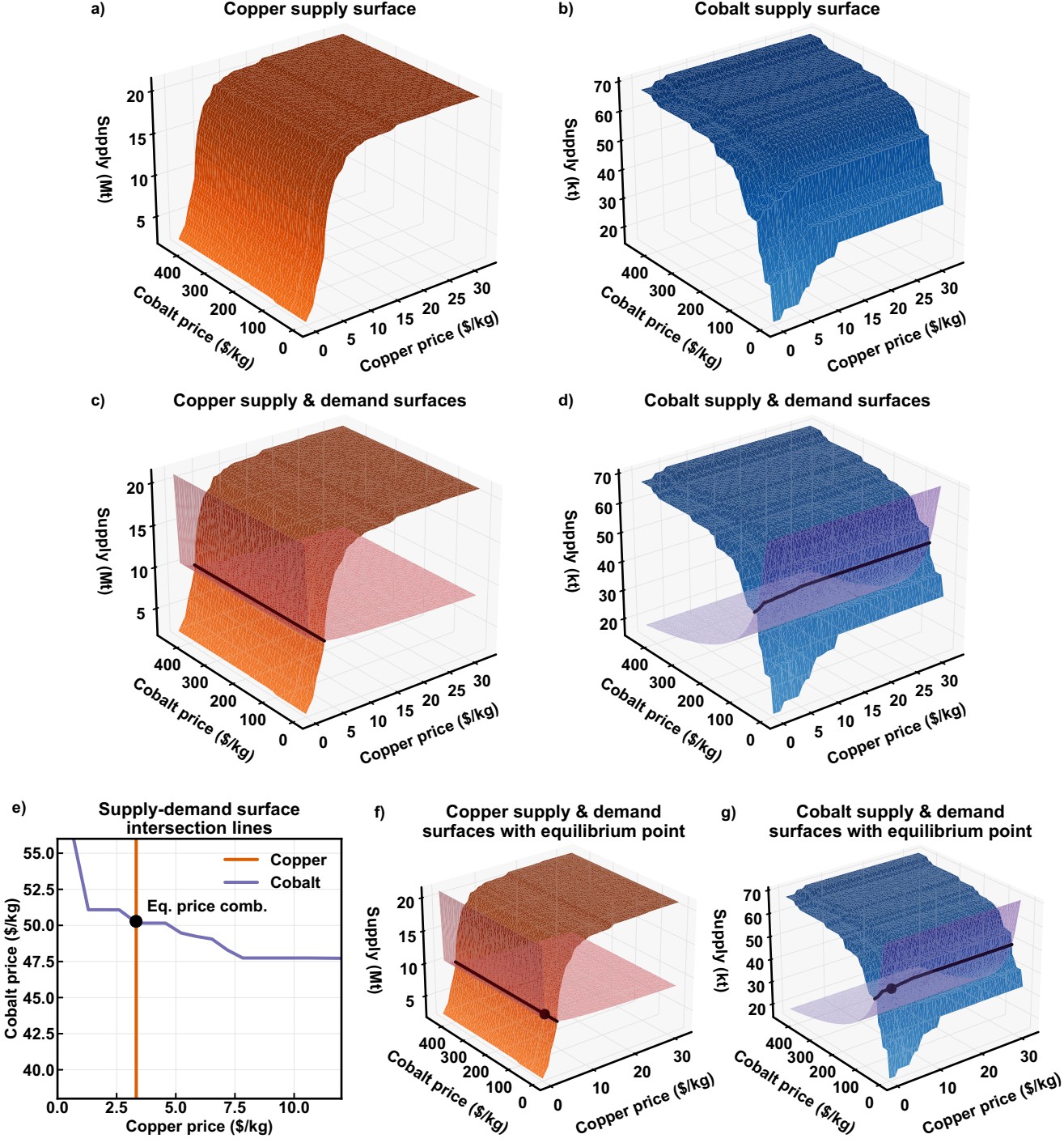

**Fig. 7 | Constructing and solving two-dimensional supply surfaces.** Supply surfaces for all mines producing copper and/or cobalt in 2001, showing the steps from constructing the initial supply surfaces for copper (**a**) and cobalt (**b**), adding the demand surfaces and drawing the intersection curve representing supply-demand equilibrium for copper (**c**) and cobalt (**d**), projecting these curves onto the supply = 0 plane where their intersection provides the price combination satisfying equilibrium for both commodities (**e**), and plotting the supply for each commodity corresponding with the equilibrium price combination for copper (**f**) and cobalt (**g**). Lighter, more transparent curves are the demand curves. kg = kilograms, kt = thousand metric tons, Mt = million metric tons.

approach and 0.2325 for the 2D approach. Equation 6 evaluates simulated price and demand proximity to historical values using the score variable, which is the sum over all commodities of the ratios between historical and simulated price and consumption. The maximum of the simulated-historical ratio and the historical-simulated ratio was used to avoid the need to rescale across commodities, as would be the case for the mean squared error or MAPE, and mitigates issues with large denominators or small numerators that would occur

if either ratio were used in isolation.

$$
\begin{aligned}
\mathrm{score}_y(D_i) = &\left( \sum_{c \in C} \max \left( \frac{\mathrm{simulated}_{\mathrm{con},y,c}}{\mathrm{historical}_{\mathrm{con},y,c}}, \frac{\mathrm{historical}_{\mathrm{con},y,c}}{\mathrm{simulated}_{\mathrm{con},y,c}} \right) \right) \\
&+ w_{\mathrm{price\ tuning}} \left( \sum_{c \in C} \max \left( \frac{\mathrm{simulated}_{\mathrm{price},y,c}}{\mathrm{historical}_{\mathrm{price},y,c}}, \frac{\mathrm{historical}_{\mathrm{price},y,c}}{\mathrm{simulated}_{\mathrm{price},y,c}} \right) \right)
\end{aligned}
$$

(6)

**Table 1 | Data sources for total demand, mine production, and secondary demand**

| | Cobalt | Copper | Nickel |
|---|---|---|---|
| Total demand | (Cobalt Institute, 2022; Darton, 2021; Project Blue, 2023; WoodMackenzie, 2022)[53,54,57,58] | (International Copper Study Group, 2018, 2021; Klose & Pauliuk, 2023)[59–61] | (Benchmark Mineral Intelligence, 2023; Elshkaki et al., 2017; International Nickel Study Group, 2021; Project Blue, 2023)[52,62–64] |
| Mine production | (Cobalt Institute, 2022; Darton, 2021; Project Blue, 2023; Roskill, 2019; Sun et al., 2019; U.S. Geological Survey, 2021, 2022)[53,57,58,65–69] | (International Copper Study Group, 2018; Klose & Pauliuk, 2023; U.S. Geological Survey, 2021, 2022)[59,61,68–70] | (International Nickel Study Group, 2021; Project Blue, 2023; U.S. Geological Survey, 2021, 2022)[52,64,68,69,71] |
| Secondary supply | (Cobalt Institute, 2022; Darton, 2021; Project Blue, 2023; Roskill, 2019; Sun et al., 2019)[53,57,58,66,67] | (International Copper Study Group, 2012, 2021; Klose & Pauliuk, 2023)[60,61,72] | (Elshkaki et al., 2017; International Nickel Study Group, 2021; Project Blue, 2023)[52,62,64] |

Where $\text{score}(D_i)_y$ is minimized for each year $y$ based on combination $i$ of demand shifters, $D_i$, for the three commodities, and $\text{simulated}_{\text{con},y,c}$, $\text{historical}_{\text{con},y,c}$, $\text{simulated}_{\text{price},y,c}$, and $\text{historical}_{\text{price},y,c}$ are the model simulated output and historical values of apparent consumption (subscript con) and price (subscript price) for year $y$, and commodity $c$.

The function $\text{score}_y(D_i)$ is expensive to evaluate and lacks known linearity, prompting the use of Bayesian optimization to minimize it[56]. With demand elasticities assumed constant[35], the method tunes the demand shifters for each commodity to minimize the error between historical and simulated price and demand in each year over 45 iterations, each consisting of three runs in parallel, as detailed in Supplementary Section 1.13.

### Reporting summary

Further information on research design is available in the Nature Portfolio Reporting Summary linked to this article.

## Data availability

The raw data generated in this study and the underlying data for each figure and table are available in the USGS ScienceBase repository accessible at https://doi.org/10.5066/P13KPFRL. The input data for mine costs and production is proprietary and is not available.

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

## Acknowledgements

This work was supported by the U.S. Geological Survey Mineral Resources Program. The authors would like to acknowledge Sara Lincoln for collecting and sharing price data for all commodities considered in this work, and Andrew Gulley for sharing cobalt mine production and timeline data.

## Author contributions

M.F., E.A., N.N., J.R., and L.B.A. formulated the initial methodological approach. J.R. developed the method for simultaneous solution of multiple supply curves and corresponding algorithms, with substantial revision aided by M.F. J.R. constructed visualizations and demand shifter projections with substantive input from K.B., M.F., E.S., R.R., E.O., E.A., and N.N. Data collection was led by J.R. and supported by E.S. and L.B.A. Data ingestion, initialization, and gap filling were performed by J.R. The historical tuning process was formulated by J.R., K.B., and E.S. The writing of the manuscript was led by J.R. and E.A., with substantive input from all other authors.

## Competing interests

The authors declare no competing interests.
