## [Transparent Peer Review file · Nature Communications]

Modeling interconnected minerals markets with multicommodity supply curves: examining the copper-cobalt-nickel system

Corresponding Author: Dr John Ryter

Version 0:

Reviewer comments:

Reviewer #1

(Remarks to the Author)

The present contribution determines market equilibrium as well as the effects of shocks in metal markets. What makes this study special is that the metals are produced in joint production, meaning that demand shocks for one metal can influence the price of another. While this question has occasionally been addressed in the literature, the contribution takes a novel methodological approach. Particularly noteworthy is the extensive data work. Although the results are presented in great detail, the contribution of the results—rather than the methods—to the literature remains unclear to me (see below).

Additionally, I have a few comments, especially regarding the methodology.

Demand Function:

- The shape of the demand curve is generally comprehensible. However, there are other variants of demand functions in the literature. A reference to the literature might be helpful here. Evan and Lewis (2002) provide a detailed overview.
- The elasticities for nickel and cobalt are taken from Shojaeddini et al. (2024). Perhaps I missed it, but where does the value for copper come from?
- For the demand shifter pathways, the same ones are used for all raw materials. While this is certainly a good initial approach, this aspect could be further discussed. Different metals exhibit different demand conditions and, consequently, different volatilities, as shown, for example, by Renner and Wellmer (2020). Thus, one could consider selecting pathways specific to each raw material.

Presentation of Results and Discussion:

- The presentation of the results places relatively little emphasis on the direction of the outcomes (see line 249 onward). The economic relationships could be briefly described in one or two sentences as the expected patterns.
- The discussion is relatively sparse in terms of references to the literature. However, there are several examples that could be cited. By way of example, Kim (2020) and Shao et al. (2020) are worth mentioning here.
- In addition to the role of joint production, there is also a connection for market equilibrium across metals through joint consumption, i.e., the combined use of metals in consumer goods. This has been described, for example, by Schischke et al. (2024) or Shammugam et al. (2019).

Methodology:

- The described methodology is, as already noted, very innovative and can account for multiple relationships. However, to my understanding, it primarily determines equilibrium for each point in the study period separately. For a given point in time, the demand shifters and cost functions of that period are incorporated. However, dynamic effects may also occur, meaning that changes in demand shifters in one period could lead to equilibrium changes for other raw materials in subsequent periods. Such dynamic effects, as modeled by Song et al. (2022), should be mentioned, for example, in line 285 onward.

- Line 344, Formula 2:

This raises questions about the interest rate and price escalation. Is the interest rate a real rate or a nominal rate? Are there risk premiums? According to Hotelling, interest and price escalation should theoretically correspond, meaning that interest and price escalation should have no influence. However, theoretical derivation is not always supported by empirical evidence. Gaudet (2007) provides an overview here.

Additionally, the question arises regarding the end of cash flow period T . Is this also defined by the size of the storage sites? As I understand the formula, only cash costs are considered, while fixed costs are excluded.

- Line 494, Formula 6:

The fundamental problem is clearly presented. However, the determination of the score is very difficult to understand.

Minor Points:

- The footer in line 200 referring to Figure 2 is confusing. It mentions "associated with 5% demand shifter growth." Perhaps I am misunderstanding, but the figures show 50% as the main scenario.
- In line 499, the description of x is somewhat unclear or possibly redundant. It appears in Formula 7 with $x=true$, as well as in the text as a function of demand.

References:

- M. Evans, A.C. and Lewis. "Is there a common metals demand curve?" *Resources Policy* 28 (3–4) (2002): 95-104.
- Gaudet, Gérard. "Natural resource economics under the rule of Hotelling." *Canadian Journal of Economics/Revue canadienne d'économie* 40.4 (2007): 1033-1059.
- Kim, Kihyung. "Jointly produced metal markets are endogenously unstable." *Resources Policy* 66 (2020): 101592.
- Renner, Sven, and Friedrich W. Wellmer. "Volatility drivers on the metal market and exposure of producing countries." *Mineral Economics* 33.3 (2020): 311-340.
- Schischke, Amelie, Patric Papenfuß, and Andreas Rathgeber. "The three co's to jointly model commodity markets: co-production, co-consumption and co-trading." *Empirical Economics* 66.2 (2024): 883-925.
- Shammugam, Shivenes, Andreas Rathgeber, and Thomas Schlegl. "Causality between metal prices: Is joint consumption a more important determinant than joint production of main and by-product metals?." *Resources Policy* 61 (2019): 49-66.
- Shao, Liuguo, Wenqin Hu, and Danhui Yang. "The price relationship between main-byproduct metals from a multiscale nonlinear Granger causality perspective." *Resources Policy* 69 (2020): 101846.
- Shojaeddini, E., Alonso, E., and Nassar, N. T. (2024). "Estimating price elasticity of demand for mineral commodities used in Lithium-ion batteries in the face of surging demand." *Resources, Conservation and Recycling*, 207, 107664.
- Song, Huiling, et al. "Dynamic dependence between main-byproduct metals and the role of clean energy market." *Energy Economics* 108 (2022): 105905.

Reviewer #2

(Remarks to the Author)

-- General Comments

This is a very professionally constructed study evaluating potential price and supply relationships and drivers in multi-commodity production systems that are common in the mining sector, where commodities are often co-produced due to their geochemical/geophysical affinities that causes accumulation within the same mineral deposits. The core analytic methods are solid and a logical extension of prior work on cost and price curves by Nguyen and others. The 2D methods appear to be quite similar to those relationships previously derived by the authors (Ryter et al., 2024; <https://doi.org/10.1111/jiec.13517>) for understanding copper, nickel, lead, zinc and gold. It would be worthwhile cross-referencing that work and noting the different emphasis and extensions beyond that in the current manuscript, as adding additional commodity 'dimensionality' and generalising this greatly strengthens this approach. The manuscript is very high quality and could be published as is without need for substantive revisions.

The commodity selection has clear relevance for the energy transition, as copper, nickel and cobalt all have strong demand growth under current scenarios for renewable energy deployment by the IEA and others. The authors also included price and ore grade data for other commodities in their calculations, using static prices (with adjustments overtime for inflation). There is very significant co-production of copper with gold, silver and molybdenum that may have greater price and revenue coupling for copper mines on average than either cobalt or nickel. So this would represent an obvious direction for further work to add additional 'dimensionality' to this analysis, and perhaps may improve explanatory power when looking at historic datasets.

The authors noted a few different avenues for improving the analysis. However, there seems to be some practical and potential data limitations to taking these analysis much further. The commercial datasets used by the study for price, cost, grade and production data of mines are the best available data sources. Expanding these datasets further for improved coverage beyond this is largely impractical given limitations with industry transparency and public disclosure in many mining jurisdictions around the world. So perhaps the authors could expand on the feasibility of these given how cost data is typically compiled, allocated, reported and disseminated for the mining sector. For instance, cost and production data for the copper sector is often reported on a copper equivalent basis (i.e. Cu-eq). Does that style of reporting hinders the analysis approach you are taking to doing this. Also there has been continual debate amongst mineral sector analysts for at least 20 years on the appropriate comparative use of C1 or C3 cash costs as you commonly see in analysis of the copper sector. This debate hasn't really made it's way into the journal literature. This has driven industry stakeholders to the gold sector to push for analysis on the basis of all-in sustaining costs (AISC), which does incorporate sustaining capital into the cost indices.

Another factor is the marginal cost of production and the optionality of investment in plant upgrades or recovery circuits for different elements at mine sites. In practice, mines can (and do) often respond to changing co-product prices through both cut-off grade optimisation in the mining operation and re-optimisation of process plant circuits to shift the relative recovery balance across commodities or trade-off throughput for recovery to maximise on price dynamics. Or to use increased prices and resource expansion due to decreased cut-off grades as motivators for adding additional process circuits or upgrades to plant throughput.

Economic and contractual relationships between mines and downstream processing facilities may also be worth considering. The authors touch on TC/RC costs, but there are complexities beyond this including potential for price

participation terms in purchase contract agreements. In the cobalt sector, there is considerable informal toll-treatment processing of cobalt ores, mediated by middle-men acting between artisanal miners and ore processing facilities. In the copper and nickel sectors, mineral concentrate pricing and smelter off-take agreements often include both payable and penalty elements, with concentration thresholds that can be set and exceeded before the element is included in pricing. The structure and conditions of concentrate purchase agreements also can create some delays in the adaptability of mineral pricing schemes to changed market conditions (although generally there is some recognition and tracking of LME and other market prices within these agreements). These types of complexities are difficult to account for in these types of studies given the limited quantitative data available, but perhaps may be worthwhile touching on briefly in the article. I

-- Specific Comments:

[Line 96-97] It is difficult to verify the claim of ~1% of cobalt was produced from cobalt-primary mines, with 73% as a 97 coproduct of copper mines and 26% as a byproduct of nickel mines (Gulley, 2023)

[Line 107] Are there any works that better theorise the "market delays" mentioned here?

[Line 166] Justify why MAPE has been selected as the measure of prediction accuracy (MAPE is sometimes frowned upon when used to evaluate time-series forecasting <https://otexts.com/fpp2/accuracy.html>).

[Line 194] Figure 2: Y-axis, are the values correct? Copper, for instance, has a 10^{94} value.

[Line 273-274] "We compare this new method with the previously most advanced method and find it offers several advantages, including better reproduction of historical price and production". This is not true for historical nickel price reproduction. See Figure 1 (f) Nickel price. MAPE is higher for 4D than for 2D.

Reviewer #3

(Remarks to the Author)

Reviewer #4

(Remarks to the Author)

(1) My major concern is the applicability of the resulting models in future. One of the important foundations for this research is that copper-cobalt-nickel share great inter-connection in the market, especially for their beneficial roles in electric vehicles and energy storage systems. However, such foundation may be undermined due to the substantial changes of this inter-connection (e.g., technological disruption). For example, it is highly possible in future that battery manufacturers shift towards low-cobalt or cobalt-free technologies such as lithium iron phosphate (LFP) batteries, while the copper and nickel still play significant roles in this field. Therefore, the model's robustness under different scenarios in future deserves a serious discussion.

(2) The authors employed Bayesian optimization for historical tuning. It is suggested to provide more details about parameter selection and optimizing processes to enhance the transparency of the method, and explain how to prevent potential overfitting to historical data, which may lead to poor predictive performance in future unseen data.

(3) The models were evaluated solely using the mean absolute percentage error (MAPE), potentially masking critical biases (e.g., insufficient fitting of extreme price fluctuations).

(4) The sub-figure numbers (a), (b),..., (e) are not marked in the graphical area in Figure 1.

Version 1:

Reviewer comments:

Reviewer #1

(Remarks to the Author)

Dear authors,
thank you for the revision. The new versions addresses all my points.
I am pleased.

Reviewer

Reviewer #2

(Remarks to the Author)

The authors provided a thorough response to all reviewer comments. As part of this, they made a number of changes to the manuscript and the supplementary information. The revised sections in both the manuscript and supplementary information read well and provide some useful additional clarifications. This represents an improvement to the academic rigor of the article, with some sentences having more clarity, some additional contextual background information and literature referred being added to the introduction, and also some minor errors being picked up (e.g. the Gulley (2023) attribution and associated claim regarding the percentage of cobalt production from copper and nickel mines).

The quantitative analysis remains solid. Although just noting that several reviewers were questioning the appropriateness of mean absolute percentage error (MAPE) for evaluating the model performance. The authors provided a rebuttal to this and their justification is acceptable. However, never hurts to see additional metrics used for measuring model performance.

As stated in the initial review, the manuscript and analysis is already very high quality and could be published in its current form. The changes made have slightly improved the quality of the article, however I would suggest there are diminishing returns to further revision.

If forced to make further recommendations, they would be to identify additional directions that could be taken to further build on this work and structure future research. Such as how to better incorporate regional trade dynamics and barriers, additional ore quality parameters or product composition restrictions/incentives (e.g. payable and penalty elements in concentrates and intermediate products), or testing of additional explanatory variables such as perceived investment risk across regions. Also it could be interesting to expand on what to actually do with this type of information to reduce supply chain criticality risks or to understand implications for trade policy.

Reviewer #3

(Remarks to the Author)

Reviewer #4

(Remarks to the Author)

The authors provided satisfactory revisions to all my comments. In my opinion, this manuscript is acceptable.

The authors would like to sincerely thank each of the reviewers for the time and effort they have put toward improving this work. It is greatly appreciated, and we believe it has strengthened the work.

Reviewer 1

Reviewer comment	Author response
The present contribution determines market equilibrium as well as the effects of shocks in metal markets. What makes this study special is that the metals are produced in joint production, meaning that demand shocks for one metal can influence the price of another. While this question has occasionally been addressed in the literature, the contribution takes a novel methodological approach. Particularly noteworthy is the extensive data work. Although the results are presented in great detail, the contribution of the results—rather than the methods—to the literature remains unclear to me (see below). Additionally, I have a few comments, especially regarding the methodology.	Thank you very much for your comments and discussion. The work is much more focused on the methodological contribution, with the results serving to demonstrate how this method’s outputs differ from and improve upon those of previous approaches. We have attempted to update the work to address your remaining comments, and hope that the additional discussion has clarified the issues.
The shape of the demand curve is generally comprehensible. However, there are other variants of demand functions in the literature. A reference to the literature might be helpful here. Evan and Lewis (2002) provide a detailed overview.	Thank you for this suggestion, we have updated the text before Equation 1 to the following: The demand curve is described by Equation 1, using the general form reflecting the inverse correlation between price and demand (Evans & Lewis, 2002), and corresponding with the derivation of the price elasticities of demand used in this work (Shojaeddini et al., 2024).
The elasticities for nickel and cobalt are taken from Shojaeddini et al. (2024). Perhaps I missed it, but where does the value for copper come from?	Thank you for catching this omission. Shojaeddini is one of the authors on this work and has calculated elasticities for several other commodities including copper for an additional publication, which is under preparation. The copper elasticity comes from this work and was derived following the same methods as those for cobalt and nickel, and the text in the supplementary information section 1.12 has been updated to reflect this information as follows: Demand elasticity values were held constant over time and were selected from those reported by Shojaeddini et al. for

	cobalt and nickel (Shojaeddini et al., 2024), with copper derived following the same method (manuscript under preparation).
For the demand shifter pathways, the same ones are used for all raw materials. While this is certainly a good initial approach, this aspect could be further discussed. Different metals exhibit different demand conditions and, consequently, different volatilities, as shown, for example, by Renner and Wellmer (2020). Thus, one could consider selecting pathways specific to each raw material.	Thank you for your comment and citation suggestion. The same demand shifter pathways were used across all commodities to enable direct inter-commodity comparisons, and with the focus of this paper being the new methodology, the complexity of producing realistic demand shifter projections was left for future work. However, we have added additional text in the discussion to describe how this could be accomplished, and included the suggested citation as part of the motivation for future work as follows: Furthermore, while this work uses simplistic commodity-agnostic demand shifter growth rates to study inter-commodity interactions, each commodity's demand is expected to grow with varying rates and volatilities (Renner & Wellmer, 2020). To enable such forecasts, demand shifters can be represented following the same approach as econometric methods for elasticity estimation, where exogenous parameters like GDP, steel production, electricity grid expansion, and electric vehicle adoption rates can be used to model demand shifter values, enabling more complex commodity-specific forecasts.
The presentation of the results places relatively little emphasis on the direction of the outcomes (see line 249 onward). The economic relationships could be briefly described in one or two sentences as the expected patterns.	Thank you for pointing out the weakness of this paragraph. It has been updated as follows: These results demonstrate a characteristic of supply and demand curve relationships across multiple commodities. Considering the demand shifter as a measure of willingness to pay, economic theory indicates that increasing the host commodity willingness to pay should increase the host commodity price and production, while also increasing byproduct commodity production and thus decreasing byproduct commodity price, all else equal. This work validates these expectations and expands on the theory above: increasing the copper demand shifter raises its demand curve, producing an increase in copper production (Error! Reference source not found.e) and an increase in copper price (Error! Reference source not found.e). This increase also raises cobalt production (Error! Reference source not found.d), but produces a decrease in cobalt price (Error! Reference source not found.d). The decreasing cobalt price is noteworthy and has not been previously explained, despite occurring in both the 2D and 4D cases. An increase in the copper demand shifter raises copper price and production, increasing copper revenues for all mines producing it and acting as a credit for copper-cobalt mines, lowering the cobalt

	price needed for positive free cash flow. All copper-cobalt mines shift lower on the cobalt supply curve, enabling higher cobalt production and lower price for the same cobalt demand curve.
The discussion is relatively sparse in terms of references to the literature. However, there are several examples that could be cited. By way of example, Kim (2020) and Shao et al. (2020) are worth mentioning here.	Thank you, these are helpful suggestions that we believe strengthen the discussion section of this work. We have incorporated the suggested references into the discussion on inter-commodity price and demand interactions as follows: Additionally, this method challenges the underlying assumptions of prior mineral supply models, namely that byproduct and coproduct commodities do not influence host commodity mine decision making. Not only does this method highlight substantial byproduct / coproduct effects on production and price for copper-cobalt and nickel-cobalt mines, but also for copper-nickel and nickel-copper mines. Game theory approaches have found that joint production enables implicit interactions even between single-commodity firms, such as copper and cobalt recyclers, due to their competition with multi-commodity firms (Kim, 2020), providing justification for these findings. Similarly, econometric studies of jointly-produced commodities' prices indicate host commodity price effects on byproduct commodity prices have been overestimated, and that byproduct price effects on host commodity prices have been underestimated (Shao et al., 2020).
In addition to the role of joint production, there is also a connection for market equilibrium across metals through joint consumption, i.e., the combined use of metals in consumer goods. This has been described, for example, by Schischke et al. (2024) or Shammugam et al. (2019).	This is an important point, and we have incorporated the suggested citations into the discussion on future work and applications of this modeling approach. We have also worked to implement cross-price elasticities of demand in this work and hope to present that work at the Extraction 2025 conference this fall. The text has been updated as follows: Future applications of this method could develop detailed forward-looking scenarios, including for other commodity systems, particularly where joint consumption or material substitution introduces cross-price elasticities of demand between commodities. Several studies have found evidence that joint consumption can be a strong driver of inter-commodity price interactions (Schischke et al., 2024; Shammugam et al., 2019), highlighting the need for explicit demand curve interdependencies.
The described methodology is, as already noted, very innovative and can account for multiple relationships. However, to my understanding, it primarily determines equilibrium for each	This is an interesting comment, thank you. We have added the following in the discussion section to address it: To enable such forecasts, demand shifters can be represented following the same approach as econometric methods for elasticity estimation, where exogenous parameters like GDP,

point in the study period separately. For a given point in time, the demand shifters and cost functions of that period are incorporated. However, dynamic effects may also occur, meaning that changes in demand shifters in one period could lead to equilibrium changes for other raw materials in subsequent periods. Such dynamic effects, as modeled by Song et al. (2022), should be mentioned, for example, in line 285 onward.	steel production, electricity grid expansion, and electric vehicle adoption rates can be used to model demand shifter values, enabling more complex commodity-specific forecasts. Spillover effects between host-byproduct pairs and their demand drivers may also need consideration to handle dynamic, multi-year impacts of demand shifter changes (Song et al., 2022).
Line 344, Formula 2: This raises questions about the interest rate and price escalation. Is the interest rate a real rate or a nominal rate? Are there risk premiums? According to Hotelling, interest and price escalation should theoretically correspond, meaning that interest and price escalation should have no influence. However, theoretical derivation is not always supported by empirical evidence. Gaudet (2007) provides an overview here. Additionally, the question arises regarding the end of cash flow period T. Is this also defined by the size of the storage sites? As I understand the formula, only cash costs are considered, while fixed costs are excluded	Thank you for these questions. As described in the discussion section, one of the limitations of this work is that we do not actually use the net present value in the model's calculations, and instead use only free cash flow in the year of interest. Real costs and prices are used throughout the study. Fixed costs / capital expenditures had been incorrectly omitted from the NPV equation, and that has been fixed. The method text has been updated to reflect this as follows: Free cash flow was used rather than NPV throughout this study for simplicity, and all costs and prices were adjusted for inflation to 2023 U.S. dollars. To address your remaining questions, there is at minimum anecdotal evidence that discount rates vary based on jurisdiction and other risk factors and are real rather than nominal rates, and the cash flow period T should be the anticipated lifetime of the mine, which would be defined by the size of the reserves and the ore processed each year. The discussion section was updated as follows: A more realistic approach would require that each new mine's net present value evaluated over its lifetime exceeds zero based on some project-dependent discount rate, which would increase the prices required for mine opening relative to the current free cash flow approach and would necessitate data on construction costs.
Line 494, Formula 6: The fundamental problem is clearly presented. However, the determination of the score is very difficult to understand. In line 499, the description of x is somewhat unclear or possibly redundant. It appears in Formula 7	This is useful feedback. We have attempted to improve the text surrounding Equation 6 to better explain the determination of the score, and have also combined Equations 6 and 7 in a way that we hope is more straightforward than the previous version.

with $x=true$, as well as in the text as a function of demand.	
The footer in line 200 referring to Figure 2 is confusing. It mentions "associated with 5% demand shifter growth." Perhaps I am misunderstanding, but the figures show 50% as the main scenario.	Thank you for catching this typo, the caption of Figure 2 should read 50% rather than 5%, and this has been corrected.
M. Evans, A.C. and Lewis. "Is there a common metals demand curve?" Resources Policy 28 (3–4) (2002): 95-104. Gaudet, Gérard. "Natural resource economics under the rule of Hotelling." Canadian Journal of Economics/Revue canadienne d'économie 40.4 (2007): 1033-1059. Kim, Kihyung. "Jointly produced metal markets are endogenously unstable." Resources Policy 66 (2020): 101592. Renner, Sven, and Friedrich W. Wellmer. "Volatility drivers on the metal market and exposure of producing countries." Mineral Economics 33.3 (2020): 311-340. Schischke, Amelie, Patric Papenfuß, and Andreas Rathgeber. "The three co's to jointly model commodity markets: co-production, co-consumption and co-trading." Empirical Economics 66.2 (2024): 883-925. Shammugam, Shivenes, Andreas Rathgeber, and Thomas Schlegl. "Causality between metal prices: Is joint consumption a more important determinant than joint production of main and by-product metals?." Resources Policy 61 (2019): 49-66. Shao, Liuguo, Wenqin Hu, and Danhui Yang. "The price relationship between main-byproduct metals from a multiscale nonlinear Granger causality perspective." Resources Policy 69 (2020): 101846. Shojaeddini, E., Alonso, E., and	Thank you for providing the full references separate from your citations above, this was quite a helpful approach.

Nassar, N. T. (2024). "Estimating price elasticity of demand for mineral commodities used in Lithium-ion batteries in the face of surging demand." Resources, Conservation and Recycling, 207, 107664. Song, Huiling, et al. "Dynamic dependence between main-byproduct metals and the role of clean energy market." Energy Economics 108 (2022): 105905.	
---	--

Reviewer 2

Reviewer comment	Author response
This is a very professionally constructed study evaluating potential price and supply relationships and drivers in multi-commodity production systems that are common in the mining sector, where commodities are often co-produced due to their geochemical/geophysical affinities that causes accumulation within the same mineral deposits. The core analytic methods are solid and a logical extension of prior work on cost and price curves by Nguyen and others. The 2D methods appear to be quite similar to those relationships previously derived by the authors (Ryter et al., 2024; https://doi.org/10.1111/jiec.13517) for understanding copper, nickel, lead, zinc and gold. It would be worthwhile cross-referencing that work and noting the different emphasis and extensions beyond that in the current manuscript, as adding additional commodity 'dimensionality' and generalising this greatly strengthens this approach. The manuscript is very high quality and could be published as is without need for substantive revisions.	Thank you very much for your feedback and kind words. To address the differences between this and previous work, the following paragraph has been added to the introduction: The authors have previously attempted generalized mineral supply chain modeling for host commodities in isolation (Ryter et al., 2024). In place of a supply curve-based approach, price-responsive reserve to production ratios were used to generate simulated mines available for opening, and the probability of mine opening was tuned to match historical production. Where this previous work was used to understand how elasticities and supply chain response rates differ across host mineral commodities, the method presented in this work could permit more robust forecasting informed by multiple future policy scenarios with improved model stability. The novel method also reduces data requirements and the model complexity per commodity while enabling the representation of multiple commodities simultaneously.
The commodity selection has clear relevance for the energy transition, as copper, nickel and cobalt all have strong demand growth under current scenarios for renewable energy deployment by the IEA and others. The authors also included price and ore grade data for other commodities in their calculations, using static prices (with adjustments overtime for inflation). There is very significant co-production of copper with gold, silver and molybdenum that may have greater price and revenue coupling for copper mines on average than either cobalt or nickel. So this would represent an obvious direction for further work to add additional 'dimensionality' to this analysis, and perhaps may improve explanatory power when looking at historic datasets.	Thank you for this important point, which fits well with the discussion of nickel's higher price responsiveness relative to copper, as follows: Similarly, although this work accounts for revenue contributions of other jointly produced commodities at all mines using static prices and production, the presence of significant gold, silver, molybdenum, and other commodity production at copper mines may dampen their price responsiveness relative to nickel. Expanding model dimensionality to account for these commodities is a potential avenue for future work.
The authors noted a few different avenues for improving the analysis. However, there seems to be some practical and potential data limitations to taking these analysis much further. The commercial datasets used by the study for price, cost, grade and production data of mines are the	These are all excellent points, thank you. There are definitely limitations to the data, though we are increasingly seeing efforts to address them, including public-private partnerships (e.g. US Dept. of Defense's DARPA OPEN program) and finding ways to fill gaps using cost modeling. The

best available data sources. Expanding these datasets further for improved coverage beyond this is largely impractical given limitations with industry transparency and public disclosure in many mining jurisdictions around the world. So perhaps the authors could expand on the feasibility of these given how cost data is typically compiled, allocated, reported and disseminated for the mining sector. For instance, cost and production data for the copper sector is often reported on a copper equivalent basis (i.e. Cu-eq). Does that style of reporting hinder the analysis approach you are taking to doing this. Also there has been continual debate amongst mineral sector analysts for at least 20 years on the appropriate comparative use of C1 or C3 cash costs as you commonly see in analysis of the copper sector. This debate hasn't really made it's way into the journal literature. This has driven industry stakeholders to the gold sector to push for analysis on the basis of all-in sustaining costs (AISC), which does incorporate sustaining capital into the cost indices.	model can also be updated to use more simplistic data such as capital expenditures and operating expenditures that are more readily obtained from company reports and filings. The method's focus on total costs makes it straightforward to use copper-equivalent or similar costs, but the C1/C3/AISC decision remains a challenge. We have added a paragraph in the discussion section that attempts to address these issues, alongside a few of the issues you've mentioned in comments below, since they can generally be summarized as the further difficulties of modeling mine production, particularly given that mines are not static entities. We have tried to briefly discuss the difficulties of modeling mines' decision making around recovery re-optimization, plant upgrading investments, and the behavior of refineries, as follows:
Another factor is the marginal cost of production and the optionality of investment in plant upgrades or recovery circuits for different elements at mine sites. In practice, mines can (and do) often respond to changing co-product prices through both cut-off grade optimisation in the mining operation and re-optimisation of process plant circuits to shift the relative recovery balance across commodities or trade-off throughput for recovery to maximise on price dynamics. Or to use increased prices and resource expansion due to decreased cut-off grades as motivators for adding additional process circuits or upgrades to plant throughput.	Given this model already utilizes some of the best available public and commercial datasets, there exist substantial challenges to adding additional cost details from a data perspective, particularly given varying transparency and disclosure practices throughout the industry. Investments in plant upgrades and recovery circuits, alongside actions that change joint production ratios and marginal costs such as process plant re-optimization, are challenging from both data and modeling perspectives. However, this method's implementation, where mine production is calculated as a function of each price combination, would be compatible with more complex mine production optimization algorithms, if data were available. The method's use of mines' total annual costs rather than commodity-level cost allocations can also permit usage of the most fundamental data reported in company filings such as annual capital and operating expenditures, negating the need to differentiate between C1 or C3 costs, or e.g. copper-equivalent costs. Similarly, mineral concentrate prices, treatment charges, and refining charges all vary with the prevalence of long-term smelting and refining contracts, as well as the presence of and charges for payable and penalty elements. While the current approach contains static representations of treatment
Economic and contractual relationships between mines and downstream processing facilities may also be worth considering. The authors touch on TC/RC costs, but there are complexities beyond this including potential for price participation terms in purchase contract agreements. In the cobalt sector, there is considerable informal toll-treatment processing of cobalt ores, mediated by middle-men acting between artisanal miners and ore processing facilities. In the copper and nickel sectors, mineral concentrate pricing and smelter	

off-take agreements often include both payable and penalty elements, with concentration thresholds that can be set and exceeded before the element is included in pricing. The structure and conditions of concentrate purchase agreements also can create some delays in the adaptability of mineral pricing schemes to changed market conditions (although generally their is some recognition and tracking of LME and other market prices within these agreements). These types of complexities are difficult to account for in these types of studies given the limited quantitative data available, but perhaps may be worthwhile touching on briefly in the article.	charges and refining charges (TCRCs), TCRCs can vary across commodities and respond to supply-demand imbalances in the mineral concentrate market (Ryter et al., 2022). Refining also has the capacity to act as a bottleneck in mineral supply (Ali et al., 2022), and representing it explicitly would improve the model.
[Line 96-97] It is difficult to verify the claim of ~1% of cobalt was produced from cobalt-primary mines, with 73% as a 97 coproduct of copper mines and 26% as a byproduct of nickel mines (Gulley, 2023)	Thank you for catching this error. The data for verifying these data is in Supplementary Table 2 of the following work, which was sent to me by the author (Gulley), and I incorrectly identified the corresponding journal. The reference has been correctly updated in the publication, and the percentages were updated slightly because I had previously only included the DRC among copper-coproduct sources, rather than all cobalt produced at copper-primary mines. https://link.springer.com/article/10.1007/s13563-024-00447-w
[Line 107] Are there any works that better theorise the “market delays” mentioned here?	Thank you for pointing this out – there is a relatively new paper that studies these market delays quite explicitly, and we have added that citation to the work after the statement referenced here (Buarque et al., 2024).
[Line 166] Justify why MAPE has been selected as the measure of prediction accuracy (MAPE is sometimes frowned upon when used to evaluate time-series forecasting https://otexts.com/fpp2/accuracy.html).	MAPE was selected due to the use of percentage errors in the Nguyen et al. 2021 work which we were comparing with this work, as well as its ease of interpretability and accessibility given Nature Communications serves a more general audience, and the different scales of the commodities’ price and production observed in this work. With no zero values for price or production, the most substantial pitfall of the MAPE metric was avoided. Other metrics such as the mean absolute scaled error (MASE), which avoids the potential issues of MAPE, are more difficult to interpret and were avoided.
[Line 194] Figure 2: Y-axis, are the values correct? Copper, for instance, has a 10^94 value.	Yes, the values are correct, and become rather extreme due to the small demand elasticities. The demand curve equation is:

	Price = $\alpha * \text{demand}^{1/\beta}$ The elasticity ($\beta$) for copper is -0.05, giving $1/\beta = -20$. Raising the demand value to the negative 20th power means that very large α values are needed to produce reasonable price values. The following sentence has been added to the results section where the demand shifters are discussed: Demand shifters are large due to small price elasticities of demand for copper (-0.05), nickel (-0.09), and cobalt (-0.45) and the formulation of Equation 1
[Line 273-274] “We compare this new method with the previously most advanced method and find it offers several advantages, including better reproduction of historical price and production”. This is not true for historical nickel price reproduction. See Figure 1 (f) Nickel price. Mape is higher for 4D than for 2D.	Thank you for pointing this out, the language here was imprecise, and we have added “for nearly all cases” to the end of this sentence.

Reviewer 3

Reviewer comment	Author response
I co-reviewed this manuscript with one of the reviewers who provided the listed reports. This is part of the Nature Communications initiative to facilitate training in peer review and to provide appropriate recognition for Early Career Researchers who co-review manuscripts.	Thank you for your time and this sounds like a good program, I hope this was useful for you and your career.

Reviewer 4

Reviewer comment	Author response
My major concern is the applicability of the resulting models in future. One of the important foundations for this research is that copper-cobalt-nickel share great inter-connection in the market, especially for their beneficial roles in electric vehicles and energy storage systems. However, such foundation may be undermined due to the substantial changes of this inter-connection (e.g., technological disruption). For example, it is highly possible in future that battery manufacturers shift towards low-cobalt or cobalt-free technologies such as lithium iron phosphate (LFP) batteries, while the copper and	Thank you for your comments, and we very much agree that making forecasts with this model deserves much more serious discussion, as well as more serious study. The demand scenarios used in this work were used only to demonstrate the model capabilities and the resulting interactions between commodities, and we have added substantial language in the discussion section describing how realistic forward-looking scenarios would be implemented, as follows: Future applications of this method could develop detailed forward-looking scenarios, including for

nickel still play significant roles in this field. Therefore, the model's robustness under different scenarios in future deserves a serious discussion.	other commodity systems, particularly where joint consumption or material substitution introduces cross-price elasticities of demand between commodities. Several studies have found evidence that joint consumption can be a strong driver of inter-commodity price interactions (Schischke et al., 2024; Shammugam et al., 2019), highlighting the need for explicit demand curve interdependencies. Furthermore, while this work uses simplistic commodity-agnostic demand shifter growth rates to study inter-commodity interactions, each commodity's demand is expected to grow with varying rates and volatilities (Renner & Wellmer, 2020). To enable such forecasts, demand shifters can be represented following the same approach as econometric methods for elasticity estimation, where exogenous parameters like GDP, steel production, electricity grid expansion, and electric vehicle adoption rates can be used to model demand shifter values, enabling more complex commodity-specific forecasts. Spillover effects between host-byproduct pairs and their demand drivers may also need consideration to handle dynamic, multi-year impacts of demand shifter changes (Song et al., 2022).
The authors employed Bayesian optimization for historical tuning. It is suggested to provide more details about parameter selection and optimizing processes to enhance the transparency of the method, and explain how to prevent potential overfitting to historical data, which may lead to poor predictive performance in future unseen data.	Thank you for bringing up these potential problems. The Bayesian optimization was only used to fit historical values of the demand shifter, with separate Bayesian optimization models initialized to minimize the error between the simulated and historical data independently in each year. Because the Bayesian optimization requires historical price and production values to operate, the optimization cannot be performed in a way that would permit predictions on future unseen data. We have added discussion about using the overall model to perform forward-looking predictions as described above, and future work will explore time-dependent train-test splits using the demand shifters as the independent variables. Supplementary Section 1.13 has also been updated to include more details about the parameter selection, as follows: The Bayesian optimization model is implemented using the scikit-optimize Optimizer class in Python (Head, 2021), with Latin hypercube initialization

	and a gradient boosted regression trees surrogate model, with the acquisition parameters κ and ξ set to 1 to balance exploration with exploitation. This implementation uses default or near-default values from the scikit-optimize package, with defaults updated only when the initial implementation performed poorly. No hyperparameter optimization was performed due to the relative simplicity of the optimization problem. Because the optimization was performed within each year individually using separate models and could not be performed on forward-looking data, model overfitting was not considered problematic.
The models were evaluated solely using the mean absolute percentage error (MAPE), potentially masking critical biases (e.g., insufficient fitting of extreme price fluctuations).	MAPE was selected due to its ease of interpretability and accessibility given Nature Communications serves a more general audience, and the different scales of the commodities' price and production observed in this work. With no zero (or small) values for historical price or production, the issues around extreme price fluctuations were avoided. Other metrics such as the mean absolute scaled error (MASE), which generally avoids the potential issues of MAPE, are more difficult to interpret and were avoided.
The sub-figure numbers (a), (b),..., (e) are not marked in the graphical area in Figure 1.	Thank you for catching this omission, it has been corrected.

Reviewer 1

Reviewer comments	Response to reviewer
Dear authors, thank you for the revision. The new versions addresses all my points. I am pleased.	Thank you very much for your comments and effort toward improving this work.

Reviewer 2

Reviewer comments	Response to reviewer
The authors provided a thorough response to all reviewer comments. As part of this, they made a number of changes to the manuscript and the supplementary information. The revised sections in both the manuscript and supplementary information read well and provide some useful additional clarifications. This represents an improvement to the academic rigor of the article, with some sentences having more clarity, some additional contextual background information and literature referred being added to the introduction, and also some minor errors being picked up (e.g. the Gulley (2023) attribution and associated claim regarding the percentage of cobalt production from copper and nickel mines. The quantitative analysis remains solid. Although just noting that several reviewers were questioning the appropriateness of mean absolute percentage error (MAPE) for evaluating the model performance. The authors provided a rebuttal to this and their justification is acceptable. However, never hurts to see additional metrics used for measuring model performance. As stated in the initial review, the manuscript and analysis is already very high quality and could be published in its current form. The changes made have slightly improved the quality of the article, however I would suggest there are diminishing returns to further revision.	Thank you very much for your comments and effort toward improving this work.
If forced to make further recommendations, they would be to identify additional directions that could be taken to further build on this work and structure future research. Such as	Thank you for these recommendations, we have added additional detail on future work to the end of the last paragraph in the Discussion section.

how to better incorporate regional trade dynamics and barriers, additional ore quality parameters or product composition restrictions/incentives (e.g. payable and penalty elements in concentrates and intermediate products), or testing of additional explanatory variables such as perceived investment risk across regions. Also it could be interesting to expand on what to actually do with this type of information to reduce supply chain criticality risks or to understand implications for trade policy.	
--	--

Reviewer 3

Reviewer comments	Response to reviewer
I co-reviewed this manuscript with one of the reviewers who provided the listed reports. This is part of the Nature Communications initiative to facilitate training in peer review and to provide appropriate recognition for Early Career Researchers who co-review manuscripts.	Thank you very much for your comments and effort toward improving this work.

Reviewer 4

Reviewer comments	Response to reviewer
The authors provided satisfactory revisions to all my comments. In my opinion, this manuscript is acceptable.	Thank you very much for your comments and effort toward improving this work.